# Salmon Skin Acid-Soluble Collagen Produced by a Simplified Recovery Process: Yield, Compositions, and Molecular Characteristics

**Krisana Nilsuwan** [1], **Umesh Patil** [1], **Chuanhai Tu** [2], **Bin Zhang** [2] and **Soottawat Benjakul** [1,*]

[1] International Center of Excellence in Seafood Science and Innovation (ICE-SSI), Faculty of Agro-Industry, Prince of Songkla University, Hat Yai, Songkhla 90110, Thailand
[2] College of Food and Pharmacy, Zhejiang Ocean University, Zhoushan 316022, China
* Correspondence: soottawat.b@psu.ac.th; Tel.: +66-7428-6334

**Abstract:** Acid-soluble collagen (ASC) is generally extracted by acid solubilization, followed by precipitation and dialysis. Such a process is typically time consuming and tedious, especially for dialysis. A simplified recovery process based on water washing/centrifugation of collagen pellets to replace dialysis was successfully developed. An ASC pellet from salmon (*Oncorhynchus nerka*) skin was obtained by salt precipitation (2.6 M). The pellet was washed with 50 volumes of distilled water (DW) and centrifuged for 0–3 cycles before lyophilization. As the washing cycles augmented, decreases ($p < 0.05$) in the NaCl content with a coincidental increase ($p < 0.05$) in the hydroxyproline content were attained. Similar protein patterns between all of the ASC samples, regardless of washing cycles, were noticeable. All of the ASCs were classified as type I collagen. FTIR spectra revealed that ASC possessed a triple helical structure with sufficient washing cycles. ASC washed with DW for three cycles (ASC-3C) was selected and characterized. ASC-3C showed high extraction yield (36.73%) and had high lightness. It exhibited high thermal stability ($T_{max}$ = 37 °C) and had an ordered phase structure. Glycine and imino acids were the dominant amino acids in ASC-3C. Therefore, a simplified recovery process could be adopted for ASC production, in which the shorter time was used without adverse effects toward ASC.

**Keywords:** salmon skin; acid-soluble collagen; washing process; characterization

## 1. Introduction

Salmon is globally popular fish associated with its delicacy and attractive orange color. Salmon meat is composed of high proteins rich in essential amino acids [1]. During processing, skins (about 7%) are removed and disposed [1]. Skins are gaining attention for conversion to collagen or gelatin or hydrolyzed collagen [2,3]. Collagen constitutes around 30% of total protein in animals. Twenty-nine types of collagens possessing varying structure and molecular properties have been documented [4]. Type I collagen is abundant in fish and mammals. Type I collagen is triple helical in structure, in which hydrogen bonds between the glycine and amide groups play a role in the stabilization of the structure [5]. Nowadays, collagens from fish processing byproducts have gained attention as a replacement for those from mammals since they can be used without religious constraint and risk from certain contagious diseases from land animals [6,7].

Typically, acid-soluble collagen (ASC) can be produced from fish skin. Process consists of skin preparation, acid extraction, salt precipitation, dialysis and lyophilization [8]. Pretreatments with alkaline solutions such as sodium hydroxide (NaOH) and sodium chloride (NaCl) have been implemented to remove non-collagenous proteins and pigments [9]. Collagen extracted using acid was basically recovered by salt precipitation, centrifugation, dialysis, and lyophilization. The NaCl at a concentration of 2.6 M with pH 7.5 was mostly used for the recovery of collagen [10–12]. To remove NaCl in collagen pellets, dialysis

against diluted acetic acid and distilled water (DW) for more than 3 days is carried out practically [13,14]. ASCs extracted from African catfish (*Clarias gariepinus*), salmon (*Salmon salar*) and Baltic cod (*Gadus morhua*) with 0.5 M acetic acid were subjected to dialysis with distilled water for 7 days at 6 °C [15]. Overall, such a typical method is time consuming associated with long time dialysis for salt removal. Therefore, new potential extraction and recovery method is needed. Basically, the collagen is water-insoluble or cannot be solubilized at a neutral pH [14]. The dispersion of precipitated collagen in DW could dissolve NaCl and the insoluble collagen could be collected after centrifugation. However, the washing/centrifugation cycles might affect the salt content and properties of the resulting collagen. However, no information concerning the use of a simplified recovery process for collagen exists. This study aimed to elucidate the impact of a simplified recovery process via washing/centrifugation for different cycles on NaCl removal and the molecular characteristics of ASC from salmon skin.

## 2. Materials and Methods

### 2.1. Chemicals

Acetic acid was obtained from Merck (Darmstadt, Germany). Sodium dodecyl sulphate (SDS), $N,N,N',N'$-tetramethylethylenediamine and Coomassie Blue R-250 were obtained from Bio-Rad Laboratories (Hercules, CA, USA). Protein molecular weight (MW) markers were procured from Sigma Chemicals (St. Louis, MO, USA). Sodium chloride and sodium hydroxide were purchased from KemAus™ (Phakanong, Bangkok, Thailand). All of the chemicals used were of analytical grade.

### 2.2. Preparation of Salmon Skin

Sockeye salmon (*Oncorhynchus nerka*) skins from the Nissui (Thailand) Co., Ltd., Songkhla, Thailand, were stored in a polystyrene box using a skin/ice ratio of 1:2. When the skins reached the laboratory, the remaining meat and scales were removed from the skins manually and washed using cold distilled water (≤10 °C). The skins were then stacked and stored at −20 °C in a polyethylene bag (for less than 1 month).

The frozen skins were reduced to small pieces ($1.5 \times 1.5$ cm$^2$) with the aid of an electric sawing machine. Non-collagenous proteins and pigments were removed from the prepared skins [16], followed by swelling in 0.05 M citric acid, prior to defatting using 30% ($v/v$) isopropanol. Thereafter, the skins were washed with distilled water (DW) before collagen extraction [14]. All of the details are shown in Figure 1.

### 2.3. Collagen Extraction and Precipitation

ASC was extracted from the swollen skins using 50 volumes of 0.5 M acetic acid for 48 h 4 °C [13]. After filtration, the filtrate was adjusted to pH 7.5 and salt at 2.6 M NaCl was added. After 1 h, a pellet containing ASC was collected by centrifugation ($10,000 \times g$ for 5 min at 4 °C). All of the details are given in Figure 1.

### 2.4. Study on Washing/Centrifugation Cycles on Recovery of ASC

The pellet was washed with DW for varying cycles (0, 1, 2 and 3 cycles). The pellet was resuspended in DW with a pellet/DW ratio of 1:50 ($w/v$) and homogenized at 5000 rpm for 30 s. The mixture was further stirred at 150 rpm for 30 min at 4 °C. After 30 min, the precipitate was centrifuged ($10,000 \times g$, 5 min, 4 °C) using a high-speed refrigerated centrifuge. After the washing process for 0–3 cycles, the pellets were collected and freeze-dried (model CoolSafe 55 ScanLaf A/S, Lynge, Denmark). ASC powders obtained from washing cycles of 0, 1, 2 and 3 cycles were named as 'ASC-0C, ASC-1C, ASC-2C and ASC-3C', respectively. All of the ASC samples were then analyzed.

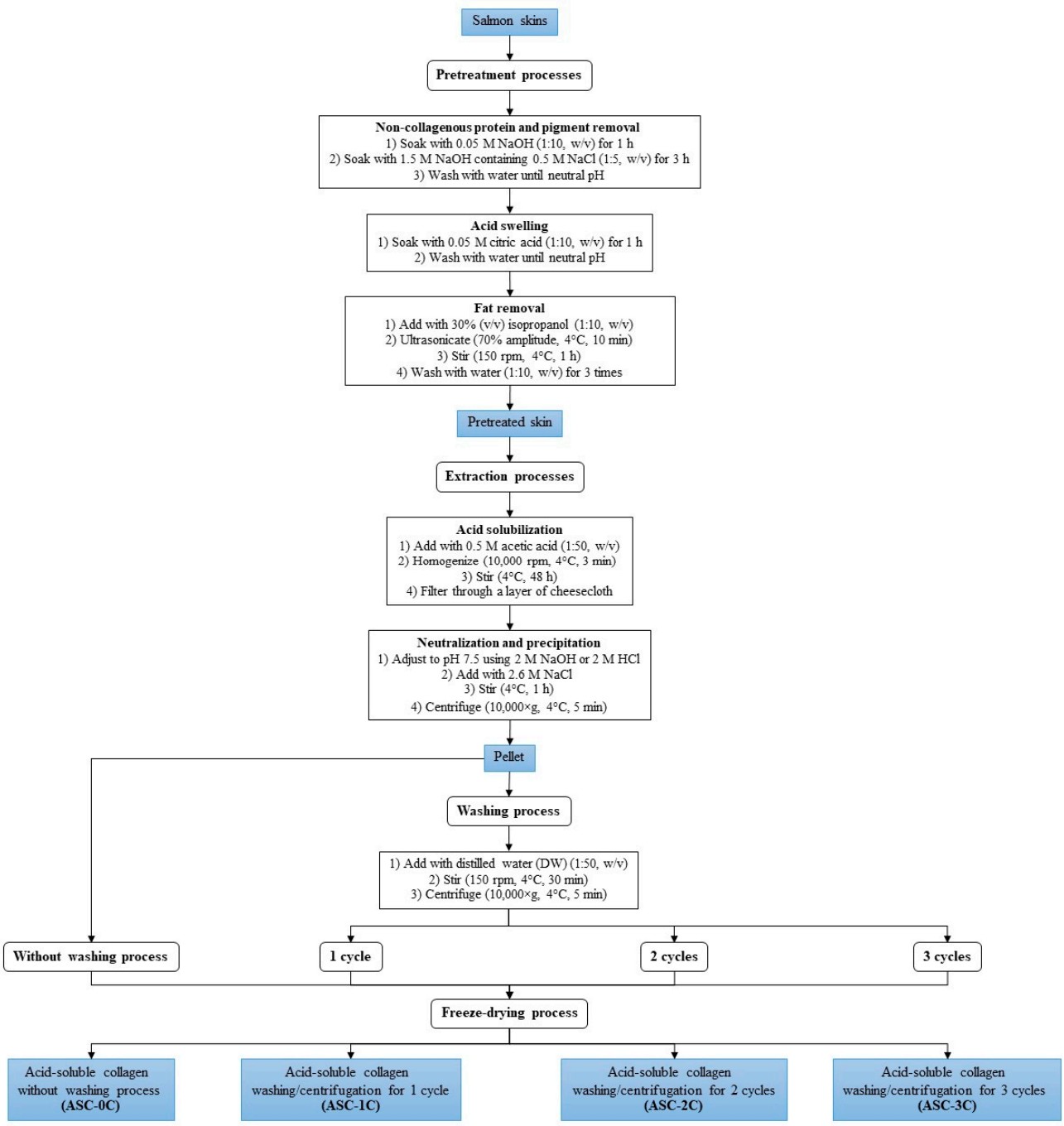

**Figure 1.** Schematic diagram for simplified recovery method of acid-soluble collagen from salmon skin.

*2.5. Analyses*

2.5.1. Sodium Chloride Content

The sodium chloride (NaCl) content of the ASC powders was determined using AgNO$_3$ according to AOAC (2001) as modified by Pongsetkul, et al. [17].

2.5.2. Hydroxyproline Content

The hydroxyproline (Hyp) content in the ASC powders was examined by a spectrophotometric method as tailored by Bergman and Loxley [18].

### 2.5.3. Protein Pattern

The protein patterns of the ASC powders were analyzed using sodium dodecyl sulphate–polyacrylamide gel electrophoresis (SDS–PAGE) [19]. The samples (15 µg protein) as quantified by the Biuret method were loaded onto polyacrylamide gel comprising 4% stacking and 7.5% running gel. After separation using a constant current of 15 mA/plate, the gel was stained and destained. The intensity of the protein bands was quantified with the aid of public domain digital analysis software (ImageJ 1.42q, National Institutes of Health (NIH), Bethesda, MD, USA).

### 2.5.4. Fourier Transform Infrared (FTIR) Spectroscopy

Dried ASC powders were subjected to ATR-FTIR analysis using model Equinox 55 (Bruker, Ettlingen, Germany) [20]. Spectra in the wavenumber range of 650–4000 $cm^{-1}$ with a step resolution of 4 $cm^{-1}$ were acquired and OPUS 3.0 software (Bruker, Ettlingen, Germany) was used.

### 2.5.5. Characterization of the Selected ASC

The ASC prepared using the washing/centrifugation cycles exhibiting the lowest NaCl content and the highest hydroxyproline content was characterized.

#### Extraction Yield

Yield was calculated as follows:

$$\text{Yield (\%)} = \frac{\text{Collagen powder (g)}}{\text{Dried weight of initial skin (g)}} \times 100 \tag{1}$$

#### Proximate Compositions

ASC powder was determined for the protein, fat and ash contents as per the method of AOAC [21] with the analytical no. of 981.10, 948.15, and 923.03, respectively.

#### Color

Color parameters including *L** (lightness/darkness), *a** (redness/greenness) and *b** (yellowness/blueness) of the ASC powder were determined using a Hunter Lab Colorimeter [22]. *ΔE** (color difference) was also examined.

#### Differential Scanning Calorimetry (DSC)

The ASC powder was firstly rehydrated in deionized water at 1:40 (*w/v*) for 24 h at 4 °C. The analysis was completed using a differential scanning calorimeter Perkin–Elmer Model DSC7 (Norwalk, CA, USA) [20] over a temperature of 20–50 °C at the rate of 1 °C $min^{-1}$. Th endothermic peak representing the maximum transition temperature ($T_{max}$) was determined.

#### ζ-Potential

The ASC solution (0.04%) was prepared in 50 mM acetic acid (*w/v*) at 4 °C. Twenty milliliters of solution were subjected to an auto-titrator model BI-ZTU (Brookhaven Instruments Co., Holtsville, NY, USA), in which the desired pHs of solution (pH 4, 5, 6, 7 and 8) were reached using 1 M nitric acid or 1 M potassium hydroxide. The zeta potential was recorded and evaluated using a zeta potential analyzer model ZetaPALs (Brookhaven Instruments Co., Holtsville, NY, USA). pH rendering zero potential was considered as pI.

#### Amino Acid Composition

ASC powder was hydrolyzed with 4.0 M methanesulfonic acid at 110 °C for 22 h under a reduced pressure to prevent the oxidation of tryptophan. The sample was then neutralized using 3.5 M NaOH and further diluted with 0.2 M citrate buffer (pH 2.2). An

aliquot (100 μL) was injected into an amino acid analyzer (JLC-500/V AminoTac^TM, JEOL USA Inc., Peabody, MA, USA).

### 2.6. Statistical Analysis

Completely randomized design (CRD) was implemented for the whole study. The experiment and analysis were conducted in triplicate ($n = 3$). Analysis of variance (ANOVA) was completed and the differences among the samples were determined using Duncan's multiple range test at the $p < 0.05$ level. SPSS package (SPSS for windows, SPSS Inc., Chicago, IL, USA) was used for analysis.

## 3. Results and Discussion

### 3.1. Sodium Chloride Content

The sodium chloride (NaCl) contents in the ASC, in which the pellet obtained from salt precipitation was washed with DW for different cycles, are shown in Figure 2A. The NaCl content of the ASC decreased ($p < 0.05$) when the washing cycle upsurged. The lowest NaCl content ($p < 0.05$) was found in the ASC washed with DW for three cycles (0.57%, dry weight basis). NaCl was reduced by 28.59, 93.80 and 99.10% when the collagen pellet was washed with DW for one, two and three cycles, respectively, in comparison with that without washing. Thus, DW was able to remove NaCl, especially with multiple washing cycles. NaCl at a high level is commonly used for collagen precipitation [13]. Therefore, the washing of the pellet with DW for three cycles rendered the ASC powder with a negligible NaCl content and it was suitable for the production of ASC from salmon skin.

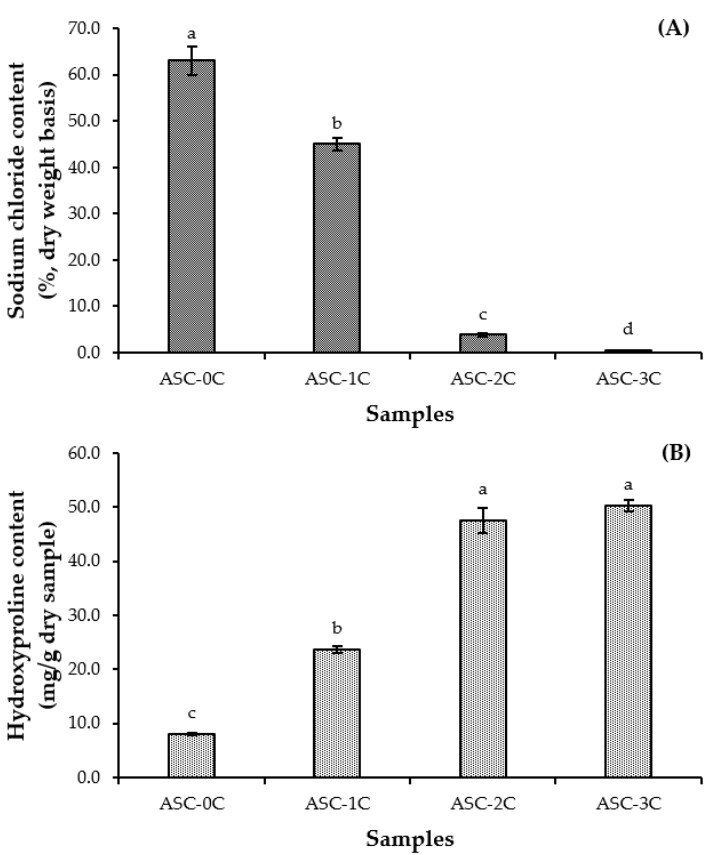

**Figure 2.** Sodium chloride content (**A**) and hydroxyproline content (**B**) of acid-soluble collagen from salmon skin prepared by washing collagen pellets using distilled water for different cycles. Different lowercase letters on the bars indicate significant differences ($p < 0.05$). Bars denote standard deviations ($n = 3$). ASC-0C, ASC-1C, ASC-2C and ASC-3C: acid-soluble collagen washed with distilled water for zero, one, two and three cycles, respectively.

### 3.2. Hydroxyproline Content

The hydroxyproline (Hyp) content of the ASC washed with DW at various cycles is depicted in Figure 2B. Theoretically, Hyp is a unique amino acid found in collagen, whereas it is not found in other proteins [9]. The augmenting Hyp content ($p < 0.05$) was attained as the cycle of washing was increased. The highest Hyp content was obtained for ASC washed with DW for more than two cycles ($p < 0.05$). The washing process for several cycles could remove NaCl more effectively, while concentrating collagen. This was evidenced by the increased Hyp content in the resulting ASC. The Hyp content of the ASC washed with DW for three cycles was 50.27 mg/g dry sample, which was similar to the Hyp content of the ASC from the salmon skin recovered with a conventional method (49.15 mg/g dry sample) [14], indicating that high purity was found for the ASC washed with DW for three cycles. Collagen is not able to solubilize in water or solution at a neutral pH. Therefore, the washing process did not leach out collagen and only NaCl was dissolved. When centrifugation was implemented, the ASC was collected as a pellet. This was advantageous for the simplified recovery process. Therefore, the number of cycles of the washing process has an important effect on the hydroxyproline content of the resulting ASC.

### 3.3. Protein Pattern

The protein patterns of the ASC prepared by washing the pellet with DW for different cycles are shown in Figure 3. All of the ASC samples comprised $\beta$-chains (dimer of $\alpha$-chains) and $\alpha$-chains ($\alpha_1$- and $\alpha_2$-chains) as major components. The $\gamma$-chain (trimer of $\alpha$-chains) was also found, suggesting that all of the ASC had high molecular weight cross-linkages [23], regardless of washing cycles. The band intensity ratio of $\alpha_1$:$\alpha_2$-chains was approximately 2:1, confirming the presence of type I collagen [14]. Type I collagen was also reported in other fish skins such as golden carp [13], bigeye tuna [24], striped catfish [25], bighead carp [26] and seabass [20]. No differences in molecular weight (MW) and intensity among all the samples were noticeable. The washing process with DW thus had no marked impact on the protein pattern of the ASC. Moreover, the remaining NaCl content in the ASC did not show any influence on the protein patterns of the resulting ASC.

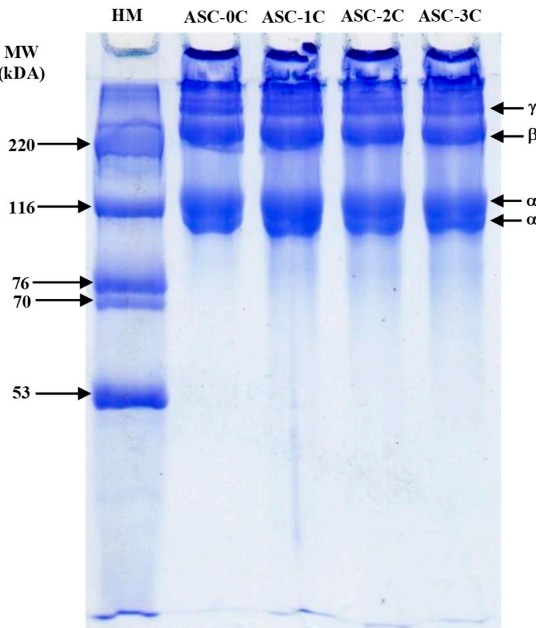

**Figure 3.** Protein pattern of acid-soluble collagen from salmon skin prepared by washing the collagen pellet using distilled water for different cycles. HM: High molecular weight marker (220, 116, 76, 70 and 53 kDa); MW: Molecular weight. $\gamma$, $\beta$, $\alpha_1$ and $\alpha_2$ denote trimer, dimer, $\alpha_1$ and $\alpha_2$ chains, respectively. For the caption, see Figure 2.

### 3.4. Fourier Transform Infrared (FTIR) Spectra

FTIR spectra of all the ASC samples are shown in Figure 4. All of the samples had similar FTIR spectra, but the amplitude varied. The collagen from the fish skin generally had the characteristic peaks, involving amide A, B, I, II and III [13]. The absorption band of amide A, associated with N−H stretching vibration, was found in the range of 3292–3296 cm$^{-1}$ [27]. All of the ASC samples had the amide A in the wavenumber range of 3292–3307 cm$^{-1}$. Asymmetrical CH$_2$ stretching is associated with the amide B band [27]. The amide B for all of the samples ranged from 2921 to 2936 cm$^{-1}$. The higher wavenumber but lower amplitude of amide A and B were attained for the ASC without washing. When the number of washing cycles increased, the amplitude upsurged. NaCl at a high concentration in the ASC powder might cause intermolecular interruption and have dilution effects on the ASC. Moreover, amide I, II and III bands were typical for collagen, appearing at wavenumbers of 1600–1700, 1500–1600, and 1200–1300 cm$^{-1}$, respectively [13]. The similar patterns in amide I, II and III bands among all of the ASC samples were observed, except the amide I and II bands of ASC-0C, which were not detected. It is well established that amide I band mainly represents the stretching vibration of C=O along the polypeptide backbone, which is a sensitive marker of polypeptide secondary structure [13]. The amide I band of ASC-1C, ASC-2C, and ASC-3C tested were noted at the wavenumbers of 1647, 1635, and 1634 cm$^{-1}$, respectively. The ASC-1C, ASC-2C and ASC-3C had the amide II region at 1556, 1528, and 1531 cm$^{-1}$, respectively. The amide II band corresponded to N−H bending [25]. ASC-3C possessed a lower wavenumber of amide I than the others, suggesting that sufficient washing could remove NaCl, which might interfere with the molecular arrangement of ASC. Simultaneously, lower NaCl content could favor higher interaction via the hydrogen bond between adjacent chains in the collagen matrix to form a triple helix. The ASC-0C sample showed the additional bands at wavenumbers of 1576 and 1410 cm$^{-1}$. Those two bands might represent some particular functional groups which could be exposed as induced by high salt. Furthermore, amide III is the combination of C−N stretching and N−H deformation, which is involved in the intermolecular interactions of collagen [20]. All of the ASC samples displayed amide III at the wavenumber of 1233–1239 cm$^{-1}$, representing the hydrogen bonds, which maintained the native structure. The triple-helical structure of collagens was substantiated by the absorption ratio between amide III and the 1450 cm$^{-1}$ bands [27]. The ratio values close to 1.0 indicated the triple-helical structure of collagen [27]. The ratio values of ASC washed with DW for zero, one, two and three cycles were 0.28, 0.79, 0.91, and 0.94, respectively. It is worth nothing that the washing process with DW could renature collagen to its native form, whereas NaCl at a high concentration adversely impacted the triple-helical structure of ASC. Additionally, the amplitude of the bands at wavenumbers of 1043, 1013, and 923 cm$^{-1}$ were evidently observed for ASC-0C, but the bands disappeared in ASC-3C.

The bands at wavenumbers around 1000 cm$^{-1}$ were related to the presence of NaCl in the sample [28]. The washing process for three cycles could therefore lower the NaCl content in the ASC and could alleviate the negative effect of NaCl used for the precipitation of collagen Moreover, it could help ASC to renature and turn to a triple-helical structure.

### 3.5. Yield and Characteristics of the Selected ASC

The ASC sample washed with DW for three cycles, which showed the lowest NaCl content along with the highest hydroxyproline content and had a triple helix structure, was selected for further characterization.

### 3.5.1. Extraction Yield

The yield of the selected ASC (ASC-3C) is shown in Table 1. The yield of the ASC sample was 36.73% (dry weight basis). The aforementioned yield was greater than the yield of ASC from the clown featherback skin (27.64%) [29] and the sea bass skin (27.77%) [30]. The yield was also higher than that of the ASC from the salmon skin using dialysis for salt removal (25.95–23.18%) [14]. This result might be associated with the differences in

preparation, extraction and recovery processes used. The use of a NaOH solution containing NaCl during pretreatment could remove non-collagenous proteins and pigments and also loosen the compactness of the skin, which could enhance the solubilization of collagen during acidic extraction. Recovery of collagen by washing the pellet to remove salt with DW for three cycles was a simple method with less time consumption for collagen production.

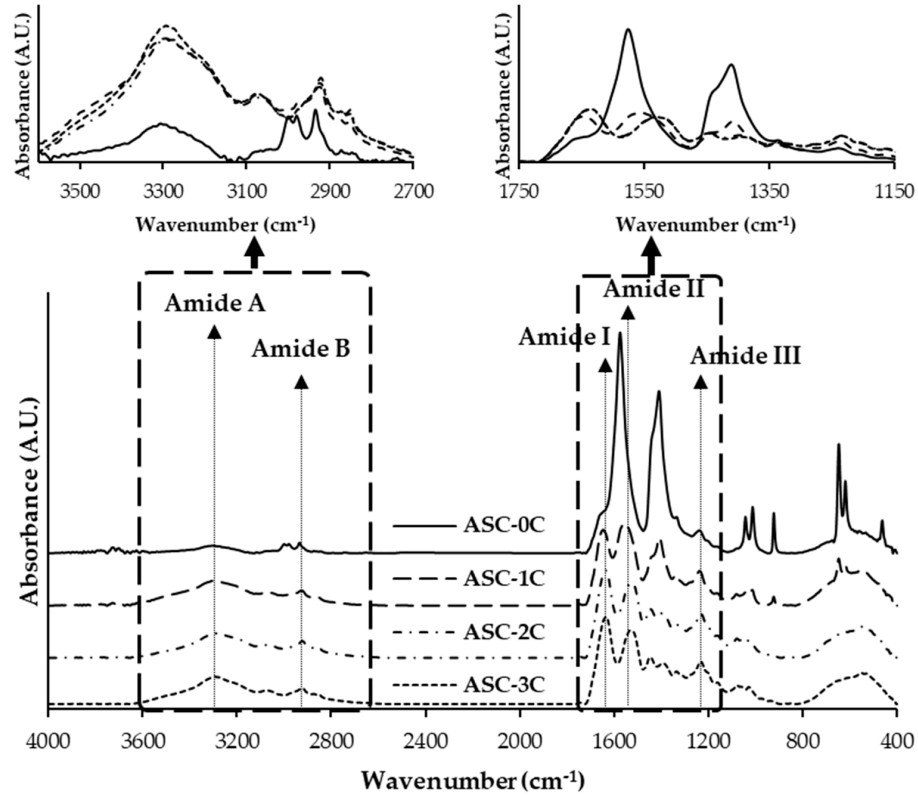

| Samples | Bands (wavenumber, cm⁻¹) | | | | | | |
|---------|---------|---------|---------|---------|---------|---------|---------|
| | Amide A | Amide B | Amide I | | Amide II | | Amide III |
| ASC-0C | 3307 | 2936 | - | 1576 | - | 1410 | 1239 |
| ASC-1C | 3294 | 2926 | 1647 | | 1556 | 1407 | 1237 |
| ASC-2C | 3292 | 2921 | 1635 | | 1528 | | 1233 |
| ASC-3C | 3292 | 2924 | 1634 | | 1531 | | 1233 |

**Figure 4.** FTIR spectra of acid-soluble collagen from salmon skin washed with distilled water at different cycles. For the caption, see Figure 2.

**Table 1.** Yield, compositions, and thermal properties of acid-soluble collagen from salmon skin prepared by washing a collagen pellet using distilled water for three cycles (ASC-3C).

| Parameters | ASC-3C |
|---|---|
| Yield (%, dry weight basis) | $36.73 \pm 3.22$ |
| Protein content (%, dry weight basis) | $97.80 \pm 0.31$ |
| Fat content (%, dry weight basis) | $1.45 \pm 0.27$ |
| Ash content (%, dry weight basis) | $0.75 \pm 0.04$ |
| $L^*$ | $82.25 \pm 0.75$ |
| $a^*$ | $0.70 \pm 0.10$ |
| $b^*$ | $3.20 \pm 0.71$ |
| $\Delta E^*$ | $11.02 \pm 0.68$ |
| $T_{onset}$ (°C) | $32.91 \pm 1.34$ |
| $T_{max}$ (°C) | $37.00 \pm 0.76$ |
| $T_{endset}$ (°C) | $41.44 \pm 0.92$ |
| $\Delta H$ (J/g) | $0.36 \pm 0.01$ |

Values are presented as mean $\pm$ SD ($n = 3$).

### 3.5.2. Proximate Analysis

The ASC-3C powder contained a high protein content (97.80%) (Table 1), indicating that the ASC became more concentrated after NaCl was removed via washing/centrifugation. Hema, et al. [31] documented that the protein contents of ASC powders from albacore tuna, dog shark, and rohu were 91.08, 88.80, and 89.94%, respectively. The protein content of ASC extracted from Nile tilapia skin with different acid concentrations and extraction times was in the range of 68.73–80.58% [32]. Moreover, the fat content of ASC-3C was very low (1.45%). Nilsuwan, Fusang, Pripatnanont and Benjakul [14] also reported the low fat content (1.86%) of ASC from salmon skin defatted with 30% isopropanol using ultrasonication. The fat contents of the ASC powder extracted from albacore tuna, dog shark, and rohu were 0.64, 0.37, and 0.33%, respectively [31]. Additionally, ASC-3C had a low ash content (0.75%). This was possibly attributed to the remaining minerals in the ASC powder. Costa, Oliveira, Droval, Marques, Fuchs, and Cardoso [32] documented that the ash content of the ASC from the Nile tilapia skin was in the range of 1.16–1.82%. The ash contents of the ASC from albacore tuna, dog shark, and rohu were 0.74, 0.76, and 0.43%, respectively [31].

### 3.5.3. Color

The color of the ASC-3C powder from the salmon skin is shown in Table 1. The $L^*$, $a^*$, $b^*$, and $\Delta E^*$ values were 82.25, 0.70, 3.20, and 11.02, respectively. ASC-3C was pale yellowish in color. The pretreatment of the salmon skin with a NaOH solution containing NaCl might promote the removal of non-collagenous proteins and pigments, which not only facilitated the extraction with high yield but also resulted in a satisfactory color for the resulting ASC powder. However, ASC-3C showed a slightly lower $L^*$ value than that of the salmon skins prepared by a typical recovery process using dialysis (85.37–85.53) as reported by Nilsuwan, Fusang, Pripatnanont, and Benjakul [14]. This might be related to the oxidation of astaxanthin during long-term dialysis, which led to a higher $L^*$-value. Nevertheless, the washing process with DW for three cycles could also improve the color of the ASC.

### 3.5.4. Differential Scanning Calorimetry (DSC)

The total denaturation enthalpy ($\Delta H$) and maximum transition temperature ($T_{max}$) of rehydrated ASC in deionized water are presented in Table 1. The $\Delta H$ value of the ASC-3C sample was 0.36 J/g. Typically, the $\Delta H$ value is related to the amount of ordered and compact structures in collagen [33]. The $T_{max}$ of ASC-3C was 37 °C (Table 1), indicating a high thermal stability. The thermal stability of the collagen was governed by the imino acid content, particularly hydroxyproline, which was involved in hydrogen bond formation [34]. High $T_{max}$ coincided with high hydroxyproline content (Figure 2B). Nevertheless, the $T_{max}$ of ASC-3C was higher than the collagen from other cold water fish including Argentine hake (10 °C) [35], Baltic cod (15.2 °C) [15], chum salmon (19.0 °C) [6] and deep-sea redfish (16.1 °C) [36] as determined by DSC. This might be related to the shorter time used for the recovery of collagen. As a result, the high integrity of the triple helix structure was still retained. Additionally, the washing process might cause the formation of collagen fibrils, resulting in a high $T_{max}$ value. A higher denaturation temperature was obtained for the collagen fibrils extracted from sturgeon skin and swim bladder, compared to those of the collagen molecules [37]. Therefore, the washing process with DW for three cycles could provide high thermal stability in the resulting ASC from the salmon skin.

### 3.5.5. ζ-Potential

The ζ-potential of ASC-3C in the pH 4–8 is depicted in Figure 5. ζ-potential has been used to monitor the changes of surface net charge of collagens at different pHs [11,12]. At the pH of 4–6, a net positive charge was observed for ASC-3C. With upsurged pHs, the positive charge of ASC was lowered. A zero net charge was found at pH 6.53, representing the pI value of ASC-3C. The pIs of ASC from the other fish skins were found at faintly acidic to neutral pHs. The result was coincidental with pI of ASC from golden carp (6.54–6.79) [13],

clown featherback (5.39–7.76) [10], and bigeye tuna (5.50–6.40) [24]. When the pH was above the pI, the surface charge turned to negative. Acidic amino acids and basic amino acids are generally involved in the charge of proteins including ASC [11].

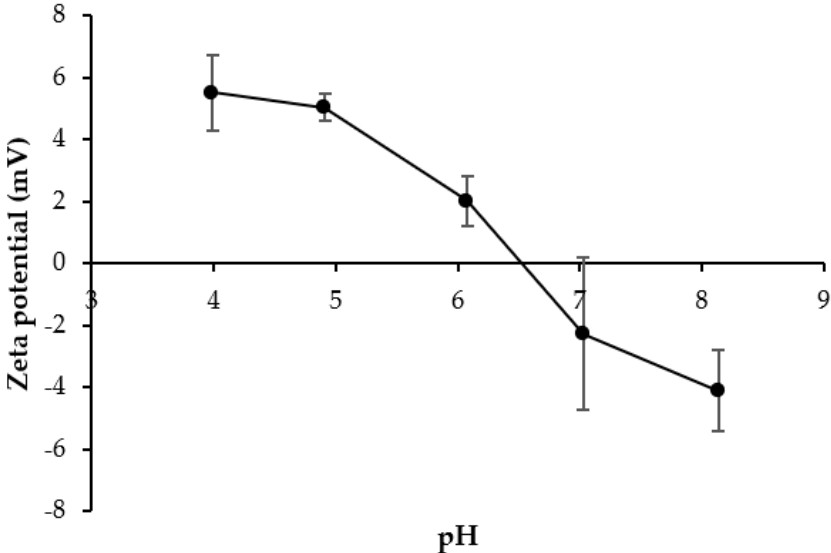

**Figure 5.** Zeta-potential of acid-soluble collagen from salmon skin prepared by washing a collagen pellet using distilled water for three cycles (ASC-3C).

3.5.6. Amino Acid Composition

Generally, amino acid compositions determine the physicochemical properties, stability and nutritive value of collagen [38]. ASC-3C had glycine as the major amino acid (302 residues per 1000 residues), followed by proline (134 residues per 1000 residues), alanine (106 residues per 1000 residues), glutamic acid/glutamine (71 residues per 1000 residues), and hydroxyproline and arginine (57 residues per 1000 residues) (Table 2). Glycine is generally localized at every third position of the α-chain (Gly-X-Y), except in the telopeptide regions [9,10]. Proline is commonly found at position X or Y, whereas hydroxyproline is detected at position Y [34]. The imino acid content (proline and hydroxyproline) is closely related to the thermal stability of collagen [9]. Furthermore, the imino acid content of the ASC-3C was in the range of 191 residues per 1000 residues, which was similar to that of the golden carp skin (194 residues per 1000 residues) but was higher than that found in the collagen from the bighead carp skin (165 residues per 1000 residues) [25,39]. The washing process with DW for three cycles yielded ASC with an amino acid composition commonly found in fish collagen.

**Table 2.** Amino acid composition of acid-soluble collagen from salmon skin prepared by washing a collagen pellet using distilled water for three cycles (ASC-3C).

| Amino Acids | Content (Residues/1000 Residues) |
|---|---|
| Alanine | 106 |
| Arginine | 57 |
| Aspartic acid/asparagine | 60 |
| Cystine | 0 |
| Glutamic acid/Glutamine | 71 |
| Glycine | 302 |
| Histidine | 1 |
| Hydroxylysine | 9 |
| Hydroxyproline (Hyp) | 57 |
| Isoleucine | 10 |

**Table 2.** *Cont.*

| Amino Acids | Content (Residues/1000 Residues) |
|---|---|
| Leucine | 20 |
| Lysine | 38 |
| Methionine | 15 |
| Phenylalanine | 17 |
| Proline (Pro) | 134 |
| Serine | 53 |
| Threonine | 26 |
| Tryptophan | 0 |
| Tyrosine | 4 |
| Valine | 19 |
| Total | 1000 |
| Imino acids (Hyp + Pro) | 191 |

## 4. Conclusions

A simplified washing/centrifugation process for three cycles is recommended to remove NaCl from precipitated collagen instead of dialysis, which takes a very long time. The ASC obtained (ASC-3C) was classified as type I collagen and it had a triple helical structure. ASC-3C had high thermal stability with an ordered structure. Glycine and imino acids were found at high levels in ASC-3C. Therefore, the above simplified recovery process could be implemented for ASC production, in which a shorter time was used for collagen manufacturing.

**Author Contributions:** Conceptualization, S.B.; methodology, S.B.; validation, S.B.; formal analysis, K.N. and U.P.; investigation, K.N.; resources, S.B.; data curation, S.B.; writing—original draft preparation, K.N.; writing—review and editing, C.T., B.Z. and S.B.; visualization, S.B.; supervision, S.B.; project administration, S.B.; funding acquisition, S.B. All authors have read and agreed to the published version of the manuscript.

**Funding:** This research was funded by the Chair Professor Grants, grant number P-20-52297 from the National Science and Technology Development Agency, Thailand.

**Institutional Review Board Statement:** Not applicable.

**Data Availability Statement:** Not applicable.

**Acknowledgments:** The support from Prince of Songkla University under the Prachayacharn program (AGR6502111N) was also acknowledged.

**Conflicts of Interest:** The authors declare no conflict of interest.

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
