# Peer review of "Salmon Skin Acid-Soluble Collagen Produced by A Simplified Recovery Process: Yield, Compositions, and Molecular Characteristics"

_fishes, doi:10.3390/fishes7060330_

Round 1
Reviewer 1 Report
In this manuscript, the authors demonstrate a simplified recovery process of acid-soluble collagen production from Salmon skin and perform molecular characterization to validate the results. The manuscript is interesting and important for researchers working in the related field.
Overall, the manuscript is written well; however, I have only concerns about the intro/background section. The authors should include more relevant papers with their results. Also, the authors should be included if any other published reports are available in the literature for acid-soluble collagen production. If yes, what is the main difference with others? Washing/centrifugation for different cycles of NaCl removal that the authors have used is common. I would like to see a revision to establish the originality of the present findings.
Author Response
Response to reviewer 1
Comments and Suggestions for Authors
In this manuscript, the authors demonstrate a simplified recovery process of acid-soluble collagen production from Salmon skin and perform molecular characterization to validate the results. The manuscript is interesting and important for researchers working in the related field.
***Thank you very much for your understanding. The reviewer’s comments and suggestions have been responded. The correction has been made as highlighted in green.
Overall, the manuscript is written well; however, I have only concerns about the intro/background section. The authors should include more relevant papers with their results. Also, the authors should be included if any other published reports are available in the literature for acid-soluble collagen production. If yes, what is the main difference with others? Washing/centrifugation for different cycles of NaCl removal that the authors have used is common. I would like to see a revision to establish the originality of the present findings.
***Thank you for reviewer’s suggestion. Actually, no information concerning the use of simplified recovery process via washing/centrifugation process for collagen production have been reported. Therefore, no relevant papers about washing/centrifugation for different cycles of NaCl removal had been provided. Typically, after precipitation of collagen using ‘salting-out’ method, the collagen pellet obtained is subjected to dialysis for several days at refrigerated temperature. The relevant reference regarding the dialysis time (7 days) for production of ASC from African catfish, salmon (Salmo salar), Baltic cod has been reported. Please see line 49 – 51 and 395 – 396.
Reference
Tylingo, R.; Mania, S.; Panek, A.; Piątek, R.; Pawłowicz, R. Isolation and characterization of acid soluble collagen from the skin of African catfish (Clarias gariepinus), salmon (Salmo salar) and baltic cod (Gadus morhua). J Biotechnol Biomater 2016, 6, 2.
To our knowledge, this is the first-time report for the development of simple method, which can be used for the large-scale production, instead of dialysis, which is costly and time consuming. However, the rationale of this study had been already provided in the introduction. Please see line 45 – 58.

Reviewer 2 Report
The authors presented a new easy mehod, which can shorten the time to obtain highly purified type I collagen from fish skin. The methods applied were appropliate, results are clear, and the conclusion is well supported from the data. In a word, this is a well-written manuscript. Only minor revisions are needed.
1) Please refere the already published reports and compare the following data with tohose of the previous reports.
a) Hyp contents of the present sample with other salmonids data. This comparison may give a concrete evidence of the purity of the present ASC sample, along with the SDS-PAGE data.
b) DSC (Tmax) data -- the reviewer feels that the denaturation temperature of the present sample is extremely high compared with other salmonid species. So, please show the already published denaturation temperature of other salmonid collagens. (When the authors compare the Tmax data, please note the method to obtain Tmax (Td), since the different methods give different value. For example, usually CD data and DSC data shows 2-3 degree difference. )
2) Discussion on the high Tmax value
The Tmax value (37 degree Celsius) of the present ASC sample seems to close to that of the "collagen fibril," not "collagen molecules." Thus, the reviewer suspect that collagen fibrils were formed during the DW-washing step of the present method. In a neutral pH solution with an appropliate salt concentration, ASC molecules are easy to form fibrils. Fibrils show higher Tmax than molecules. For example, Zhang et al. (2019, https://doi.org/10.1016/j.ijbiomac.2019.07.021) showed that sturgeon skin and swim bladder collagen fibrils' Td are 6-10 degree Celsius higher than those of molecules.
Therefore, the reviewer takes objection to the discussion that the shorten extraction time gave high Tmax (L297-299). Please reconsider the discussion on the high Tmax.
Author Response
Response to reviewer 2
Comments and Suggestions for Authors
The authors presented a new easy method, which can shorten the time to obtain highly purified type I collagen from fish skin. The methods applied were appropriate, results are clear, and the conclusion is well supported from the data. In a word, this is a well-written manuscript. Only minor revisions are needed.
***Thank you very much for your understanding and suggestions. All reviewer’s comments and suggestions have been responded and the correction have been provided in text as highlighted in yellow color.
1) Please refer the already published reports and compare the following data with those of the previous reports.
- a) Hyp contents of the present sample with other salmonids data. This comparison may give a concrete evidence of the purity of the present ASC sample, along with the SDS-PAGE data.
***Thank you very much for valuable suggestion. More discussions have been provided in text for hydroxyproline content of ASC from salmon skin reported from other works. Please see line 179 – 182.
- b) DSC (Tmax) data -- the reviewer feels that the denaturation temperature of the present sample is extremely high compared with other salmonid species. So, please show the already published denaturation temperature of other salmonid collagens. (When the authors compare the Tmax data, please note the method to obtain Tmax (Td), since the different methods give different value. For example, usually CD data and DSC data shows 2-3 degree difference. )
***Thank you for valuable comment. The information of published denaturation temperature of other salmonid collagens such as chum salmon and other fish skins had been provided and comparatively discussed in text. All data were obtained by DSC. Please see line 300 – 303.
Additionally, the method to obtain Tmax based on differential scanning calorimetry (DSC) had actually mentioned in text. Please see line 132 – 133.
2) Discussion on the high Tmax value
The Tmax value (37 degree Celsius) of the present ASC sample seems to close to that of the "collagen fibril," not "collagen molecules." Thus, the reviewer suspect that collagen fibrils were formed during the DW-washing step of the present method. In a neutral pH solution with an appropriate salt concentration, ASC molecules are easy to form fibrils. Fibrils show higher Tmax than molecules. For example, Zhang et al. (2019, https://doi.org/10.1016/j.ijbiomac.2019.07.021) showed that sturgeon skin and swim bladder collagen fibrils' Td are 6-10 degree Celsius higher than those of molecules. Therefore, the reviewer takes objection to the discussion that the shorten extraction time gave high Tmax (L297-299). Please reconsider the discussion on the high Tmax.
***Thank you for valuable suggestions. The reviewer’s suggestion is well taken into consideration and we do agree with this suggestion. More discussion and reference have been added in text. Please see line 305 – 308 and 448 – 449.

Round 2
Reviewer 1 Report
N/A